# A Phenomenological Study of Primary Healthcare Nurses’ Experiences in Assessing and Managing Diabetic Foot Complications

**DOI:** 10.3390/ijerph22121799

**Published:** 2025-11-28

**Authors:** Simiso Ntuli

**Affiliations:** Department of Podiatry, Faculty of Health Sciences, University of Johannesburg, P.O. Box 524, Auckland Park, Johannesburg 2006, South Africa; sntuli@uj.ac.za

**Keywords:** nurses, diabetic foot, primary healthcare, foot health, podiatry, chronic disease management, diabetes mellitus, foot ulcers, preventive care

## Abstract

Background: In South Africa, diabetic foot complications (DFC) often present for the first time at the primary healthcare (PHC) level, where nurses are central to diabetes management and prevention. Despite their key role, there are limited data on nurses’ experiences in managing DFC, particularly in resource-constrained primary healthcare (PHC) settings. This study’s objective was to explore and describe nurses’ experiences in managing DFC at the PHC level, with the aim of informing future interventions and policy development. Methods: A descriptive phenomenological study was conducted with 21 PHC nurses working in community healthcare centres to explore their experiences in managing DFC within PHC facilities. Results: Five key themes emerged: limited knowledge, time and workload constraints, lack of referral feedback, need for targeted training, and resource constraints. These reflect systemic and practical barriers that hinder effective care and contribute to inconsistent patient outcomes. Nurses are vital to early detection and intervention but face structural challenges that limit their capacity to deliver optimal care. Conclusions: Nurses encounter significant challenges when implementing routine diabetic foot screening at the primary healthcare level. These challenges include limited training, inadequate resources, and poor interprofessional collaboration. To overcome these barriers, targeted capacity-building initiatives, improved referral and communication pathways, and the provision of essential tools and support are needed. Additionally, integrating podiatrists into primary healthcare teams and establishing multidisciplinary foot care services are crucial steps to enhance care quality and reduce complications in resource-limited settings.

## 1. Introduction

Diabetic foot complications are among the most severe and costly outcomes of diabetes mellitus, significantly contributing to morbidity, disability, and premature mortality [1,2,3]. In South Africa (RSA), the prevalence of diabetes [4] and diabetic foot disease is increasing at an alarming rate [5]. Data from Gauteng Province indicate that 60% of diabetes-related hospital admissions are due to diabetic foot complications [6]. Studies show that diabetes-related amputations account for more than half of all lower limb amputations in the country [7,8,9]. These trends underscore the urgent need for effective prevention and early intervention, particularly at the primary healthcare (PHC) level.

PHC clinics and state hospitals in South Africa serve over 47 million uninsured people, accounting for approximately 84% of the national population [10,11]. Most patients with diabetes in South Africa first present at the PHC level, where they receive routine care. However, foot health—a critical aspect of diabetes management—is often overlooked. The lack of dedicated foot care services, routine screening, and preventive education at PHC facilities undermines early detection and contributes to avoidable complications such as ulcers and amputations. This neglect disproportionately affects vulnerable populations who rely on PHC clinics for accessible care, further widening gaps in chronic disease management.

In South Africa, PHC facilities are predominantly nurse-led and serve as the first point of contact for most citizens, especially those without private healthcare access. Nurses are central to service delivery at this level, responsible for assessing, managing, and referring patients [12,13]. However, the rising incidence of severe cases such as diabetic foot sepsis and amputations at tertiary care levels raises concerns about the effectiveness of diabetic foot care at the PHC level. Evidence suggests that nurses may lack adequate training and exposure to essential screening tools, such as the 10 g monofilament, which are critical for early detection and prevention [7,14,15]. These gaps highlight the need to explore nurses’ lived experiences in managing diabetic foot conditions to better understand frontline challenges and inform contextually relevant strategies aimed at reducing avoidable amputations.

A closer look at PHC settings reveals several systemic issues that further compromise diabetic foot care. These include inconsistent use of risk classification tools, absence of structured protocols, and delays in accessing specialist services due to weak referral systems. Moreover, diabetic foot care remains poorly integrated into routine chronic disease management, and nurses often lack formal training and ongoing professional development in this area. Together, these shortcomings contribute to fragmented care and missed opportunities for early intervention.

While prior studies have examined nurses’ knowledge and attitudes, there is limited qualitative insight into their lived experiences [16,17,18]. These local challenges mirror global concerns, where literature consistently highlights the importance of equipping nurses with the skills and resources necessary to effectively manage diabetic foot complications. International literature consistently affirms the vital role of nurses in primary healthcare settings in preventing diabetic foot complications, stressing the importance of equipping them with the skills and tools necessary for early identification, assessment, and management [19,20,21]. Without understanding the practical challenges they face, efforts to improve diabetic foot care risk being misaligned with the realities of frontline practice.

To address this gap, the study employs a phenomenological approach to explore PHC nurses’ experiences in managing diabetic foot complications. Their narratives offer insight into the practical realities of care, such as decision-making under resource constraints, professional confidence, and systemic inefficiencies—often overlooked in policy and planning. These findings could inform contextually relevant strategies, including targeted training, integration of diabetic foot care into routine PHC workflows, and improvements in referral systems and care protocols. By grounding recommendations in lived experience, the study supports policy reform and highlights the need for multidisciplinary collaboration among nurses, podiatrists, and physicians to strengthen diabetic foot management and promote more equitable, patient-centred care.

## 2. Materials and Methods

The guidelines for conducting qualitative studies established by the Consolidated Criteria for Reporting Qualitative Research (COREQ) and Standards for Reporting Qualitative Research (SRQR) were followed [22,23].

### 2.1. Design

This qualitative study employed a descriptive phenomenological design to explore the lived experiences of South African nurses in managing diabetic foot complications at the PHC level. Descriptive phenomenology is particularly valuable in areas where existing research is limited. It reveals the essence of any phenomenon under investigation by focusing on its characteristics. Phenomenology was selected for its ability to uncover the meanings that individuals assign to their experiences, particularly within complex healthcare systems.

### 2.2. Setting and Participants

The study was conducted in Community Healthcare Clinics located in Gauteng. A two-stage sampling strategy was employed to ensure representativeness and relevance. In the first stage, seven CHCs were randomly selected from a total of 38 CHCs across Gauteng’s five health districts, namely Johannesburg, Ekurhuleni, Tshwane, Sedibeng, and West Rand [24] using simple random sampling, which minimised selection bias and enhanced the generalisability of the study sites. CHCs are PHC facilities that provide similar services to clinics but operate 24/7, including emergency care and maternal services, with a capacity for short-term patient care and essential medical procedures, although not full surgical operations.

In the second stage, purposeful and maximum variation sampling was used to recruit the nurses [25]. A minimum of three nurses were recruited from each clinic. This sampling method enabled the researcher to choose participants with diverse backgrounds in terms of district location, age, gender, and years of experience, allowing for a deeper understanding of the phenomenon being studied. A minimum sample size of 12–15 participants was anticipated based on the principle of data saturation [26]. The inclusion criteria were as follows: at least 5 years of professional nursing experience, a clinical role, and willingness to share personal experiences.

While purposeful sampling enabled the collection of rich, context-specific data, it inherently limits the broader generalisability of the findings. The study was geographically restricted to selected PHC facilities within South Africa, which may not reflect the diversity of healthcare settings across the country or in other regions. Additionally, the reliance on self-reported experiences introduces the possibility of response bias, as participants may have unintentionally over- or under-emphasised certain aspects of their practice due to personal perceptions or professional pressures. Despite these limitations, efforts were made to enhance credibility by selecting participants who provided detailed insights into clinic environments, professional roles, and care delivery processes.

### 2.3. Data Collection

#### 2.3.1. Participant Recruitment Strategy

Participants were recruited from seven identified clinics using a structured approach to ensure voluntary and informed participation. The researcher visited each clinic and approached nurses directly to explain the purpose of the study and invite them to participate. To facilitate easy access to study information, a QR code was provided at each site, linking to an information letter that outlined the study’s aims, objectives, and ethical considerations. Through this platform, nurses could indicate their availability and submit their contact details, enabling the researcher to schedule interviews at a time convenient for them. This approach ensured transparency, flexibility, and participant autonomy in the recruitment process.

#### 2.3.2. Data Collection Procedure

Data were collected from 21 nurses through semi-structured interviews that lasted approximately 35–50 min and were conducted in private spaces within the participants’ clinics. Each interview began with an open-ended prompt such as: “Please share your experiences in managing patients with diabetic foot complications at your clinic.” Follow-up questions were adapted based on participants’ responses to explore key focus areas. A pilot study was conducted with three nurses from clinics not included in the main study to test the clarity and relevance of the interview guide. The pilot confirmed that all questions were easy to understand and required no modification; these participants were excluded from the main study.

The interview guide explored participants’ experiences, perceived challenges, training and preparedness, referral practices, and systemic support in managing diabetic foot complications. All interviews were conducted in English, audio-recorded with consent, and transcribed verbatim. Data saturation was achieved after approximately 18 interviews, as subsequent discussions produced recurring information without introducing new insights or themes.

### 2.4. Data Analysis

Data were analysed using a thematic approach grounded in phenomenological methodology, which seeks to uncover the essence of participants’ lived experiences. The process began with immersion in the data, involving repeated readings of interview transcripts to develop a holistic understanding of the narratives. The transcribed data were analysed using thematic analysis guided by Colaizzi’s seven-step method, which closely adheres to the principles of descriptive phenomenology [27,28,29].

The researcher followed a constant comparative analysis approach during the data analysis [22]. This approach allowed the researcher to identify similarities and differences across the dataset, and facilitated the recognition of patterns and meanings within participants’ narratives [30,31]. Before commencing analysis, the researcher engaged in bracketing by reflecting on prior knowledge and assumptions related to clinical training and supervision, supported by a reflexive diary to document observations and experiences during interviews [32]. This process minimised bias and enhanced confirmability. The analysis began with repeated readings of the transcripts to achieve immersion, followed by the extraction of significant statements and formulation of meaning units. These were clustered into initial themes, such as “resource constraints”, “emotional burden”, and “adaptation strategies”, which were iteratively refined through comparison across cases. Themes were then organised into broader categories that captured the essence of participants’ lived experiences. Atlas.ti 25 software was used to manage coding, organise data, and maintain an audit trail throughout the process.

### 2.5. Bracketing and Reflexivity

Rigid bracketing and reflexivity were applied throughout data collection, analysis, and interpretation to preserve the authenticity of these experiences. Bracketing involved systematically setting aside the researcher’s prior assumptions and theoretical position to approach the phenomenon from a neutral perspective. Reflexivity entails ongoing critical self-reflection on the researcher’s positionality and potential influence on the research process.

### 2.6. Rigour

To ensure rigour, the study adhered to Lincoln and Guba’s [33] trustworthiness criteria. Credibility was enhanced through member checking, where participants reviewed interview summaries and preliminary themes to confirm accuracy and resonance with their experiences. Five participants identified spelling mistakes, with no significant corrections made to the transcripts. Dependability was supported by maintaining a detailed audit trail of methodological decisions and using consistent interview procedures across all participants. Confirmability was achieved through reflexive journaling, which documented the researcher’s assumptions, reflections, and decision-making processes to minimise bias. Transferability was facilitated by providing thick descriptions of the research context, participant characteristics, and data collection process, enabling readers to assess applicability to other settings [34,35].

### 2.7. Triangulation

Triangulation is a qualitative research strategy that enhances validity and reliability by incorporating multiple data sources, methods, perspectives, or researchers to examine a phenomenon [36,37]. This study applied data source triangulation by including participants with varied district locations, ages, nursing qualifications, and professional experience. Such diversity enabled comparison across contexts, reducing the influence of any single demographic or geographic viewpoint. By capturing both common and contrasting experiences, triangulation strengthened the credibility and transferability of the findings.

To further support trustworthiness, the research process included reflective memos documenting analytic decisions and interpretations, contributing to the study’s auditability and confirmability. Themes were developed through an iterative and reflexive process and refined until data saturation was reached—defined as the point at which no new themes or insights emerged. Saturation occurred after approximately 18 interviews.

This rigorous approach ensured that the final themes authentically reflected the lived experiences of PHC nurses, consistent with the phenomenological aim of capturing the essence of experience across diverse contexts.

### 2.8. Ethical Considerations

Ethical approval for the study was obtained from the Faculty of Health Sciences Research Ethics Committee of the University of Johannesburg (ethics clearance number: REC-1954-2023) and the Gauteng Department of Health (ethics clearance number: NHRD REF. NO: GP_202404-075).

Each participant was assigned a number to ensure anonymity and confidentiality, and no personal identifying data were collected from them. The participants were informed of their right to withdraw from the study at any point during the data collection. They were also made aware that their data could not be removed if they chose to withdraw after the data analysis had begun, as identifying individual contributions would no longer be possible.

## 3. Results

A total of 21 nurses working in CHC clinics were recruited; 71% (n = 15) were women. Table 1 presents the participants’ characteristics.

The overall findings of the study showed that nurses at the PHC level encounter numerous challenges in assessing and managing diabetic foot complications. Excessive workloads, poor or non-existent support systems, and a lack of confidence in handling complex cases shape nurses’ experiences. These challenges are further intensified by limited knowledge, insufficient clinical training, and the absence of standardised guidelines, which collectively hinder consistent and effective care. Nurses often feel ill-equipped to manage these patients, which contributes to a negative and discouraging professional experience. Table 2 presents the five key themes and subthemes that emerged from the data. This is followed by the presentation of each theme and exemplar statements, which are presented in the narrative section, illustrating the depth and nature of the challenges faced.

### 3.1. Knowledge of Diabetic Foot Stratification and Ulcer Classification

Nurses face difficulties in diabetic foot risk stratification and ulcer classification. They did not use or understand the importance of diabetic foot stratification tools, such as the IWGDF and ADA, or wound classification systems, such as SINBAD, Wagner, or PEDIS, when managing diabetic foot complications.


*“No, we do not use any classification. There is no standardised document except for an old guideline that just says refer to a specialist, but nothing on what I, as a nurse, should do for the patient with a foot complication.”*

*[N7, N11]*



*“We just refer if the patient complains too much or the wound looks infected, but I don’t know if they must be classified.”*

*[N1, N20]*



*“No one has taught us about the need to classify, so we do the best we can and refer them to the hospital. The care of these patients depends on the resources available at the clinic. In some clinics, there are podiatrists, so patients’ feet are well looked after; in others, there’s nothing.”*

*[N2–N19]*


#### Subtheme: Clinical Knowledge Gaps

There is a lack of consensus on how primary healthcare nurses should manage diabetic foot complications. Currently, no established guidelines outline ulcer grading or provide clear indications for referring patients to higher levels of care.


*“I do not know those classifications…and I must admit I do not read much about diabetic foot complications. The guidelines we have from the central government do not explain much; they just say that you must refer if there is a wound. Therefore, the protocol is to refer upwards.”*

*[N3–N10, N13–N21]*



*“It is like we are working in the dark. You just refer. You are unsure about the severity of the wound, who will see the patient, and how soon the patient will be seen.”*

*[N2–N20]*


### 3.2. Systemic Barriers to Effective Care

Nurses confirmed that they often have limited time to conduct thorough assessments and provide patient education. In some clinics, a nurse sees up to 40 patients or more in one day due to the influx of patients in these settings. Time constraints were one area that all nurses cited as an uncontrollable variable, as the number of patients visiting their clinics kept increasing.

#### 3.2.1. Subtheme: Competing Demands in Clinical Workflow


*“Consultations are often rushed, leaving little opportunity for us to discuss foot care practices or the warning signs of complications with patients.”*

*[N2, N3, N20]*



*“There is no time at all; we see a lot of patients, so we do not check their feet.”*

*[N2–N18]*


#### 3.2.2. Subtheme: Strain from High Patient Turnover


*“We expect to see approximately 40 patients per day. If you take too long with one patient, the ones outside will start complaining and shouting, so you push.”*

*[N1, N6, N9, N21]*



*“We do not have time to wait for patients to remove their shoes and hosiery to examine their feet, there are tons of people waiting to be seen.”*

*[N2–N18]*


#### 3.2.3. Subtheme: Difficulty Prioritising Diabetic Foot Care

All participants agreed that an excessive workload negatively affects the quality of care. They felt that they were focused more on meeting quotas than on providing quality care. Although they recognised the importance of foot screenings for determining risk, these screenings were not conducted due to a lack of allocated time and insufficient skills to perform the procedure.


*“First price is diabetic control, my focus is on how well the patient is controlled, and then move on to how the other conditions are controlled as well.”*

*[N1–N21]*



*“When they report a foot problem, we just send them to the hospital, in most cases, even without seeing what is wrong with the foot”.*

*[N1, N7, N20]*



*“Time with each patient is limited, so I don’t really bother; it’s not like they only have diabetes, they also have other conditions that they need you to deal with during the consultation.”*

*[N8, N9, N13]*


### 3.3. Poor or Non-Existent Feedback After Referring

Nurses experienced poor feedback and highlighted the opportunity for improvement in communication by noting that they often did not receive enough feedback after referring patients to hospital departments. This feedback is essential for enhancing patient care and ensuring effective collaboration among healthcare providers.


*“We refer patients, but we never receive updates on treatment outcomes or changes in patient condition.” *

*[N1, N12–N19]*



*“They will go there, and someone will sort them out. I do not know; perhaps they will refer to the correct department. No one tells us anything.”*

*[N6, N18]*



*“Sometimes you feel undervalued, like your contribution does not matter. The interest you may have taken in the patient just fizzles out because you never hear about their progress.”*

*[N1, N16, N21]*



*“With the referral system and feedback, there is a big gap that we are still experiencing, even with other patients, not just patients with foot problems. This has been raised in many meetings to say, “Let us work smarter.”*

*[N19, N20]*


#### Subtheme: Breakdown in Interfacility Communication

Poor feedback was associated with poor continuity of care. Nurses felt that poor feedback was linked to poor continuity of care and compromised the overall quality of care.


*“You are expected to dress the patient’s wound with dressings that we do not even keep at this level. They refer patients back to PHC, but these patients still have to go to the hospital to get some of the dressings they need, as we don’t keep them at PHC.”*

*[N12, N21]*



*“When they discharge patients, they expect us to continue their care. No one calls and asks if you have any training or what you have in your clinic to ensure continuity of care.”*

*[N11–N21]*



*“The expectation is that you are a nurse; you must deal with it.”*

*[N3, N5, N7]*


### 3.4. Need for Targeted Training

Nurses felt inadequately prepared to manage diabetic foot complications because of limited training. Most of them had never received training on diabetic foot complications and their management.


*“Personally, I am not at all confident in diabetic foot assessments. I would like to receive training.”*

*[N1–N18]*



*“I would like to know how to stage a diabetic foot and identify the foot at risk early. With all the talk about amputations, not much is done to invest in nurses’ training on diabetic foot.”*

*[N14–N21]*



*“With a foot ulcer, we do not receive much training; you do not know the extent of the ulcer and all that. Therefore, we mostly refer to them. I am not confident at all. We need training on diabetic foot.”*

*[N1–N16]*


#### Subtheme: Limited Exposure to Diabetic Foot Care Protocols

All nurses reported low confidence in managing diabetic foot complications, primarily due to a lack of training, which directly affected timely referrals to higher levels of care.


*“We are not really confident—as nurse clinicians we have the basic training, which is a recognised qualification, but we are not confident or trained when it comes to the foot, especially the diabetic foot.”*

*[N1–N20]*



*“It would be nice to have support for foot patients, as we do for dental, you can always refer to dentistry or oral health.”*

*[N1, N9, N13]*



*“We do not receive training or education on wound care specific to patients with diabetes, and this affects our confidence when managing these patients. This makes you unsure of when, to whom, or what to refer.”*

*[N1–N20]*



*“We may be delaying these patients because we don’t know when or how soon to refer them. This could have a direct impact on why some patients get to the hospital late.”*

*[N3, N14, N17, N21]*


### 3.5. Limited Resources, Including Diagnostic Tools

Nurses recognise the vital importance of well-organised foot health services, which are most effectively provided by podiatrists. The shortage of podiatrists at primary healthcare facilities has resulted in inadequate support for addressing complex diabetic foot issues, thereby impeding effective care.


*“I understand that podiatrists are trained in South Africa, but they are rarely found in PHC. My clinic does not have any available.”*

*[N1, N11–N20]*



*“Even when podiatrists are accessible, they are often absent from many clinics. For some patients, reaching these distant clinics can be quite challenging.”*

*[N2–N13]*



*“There are no foot health services in South Africa, unlike oral, eye, and mental health services. As a nurse, you have to manage foot complications on your own, as there is no support or guidance for you in dealing with complex cases.”*

*[N2–N17]*



*“We need them for support, sometimes for simple things like cutting corns in the patients’ feet.”*

*[N1, N14–N21]*


#### Subtheme: Limited Access to Assessment Technologies

Nurses reported a lack of knowledge and a shortage of essential equipment for diabetic assessments, including Neurotips, 10 g monofilaments, and tuning forks. The unavailability of this necessary equipment can put patients at risk of injury, as nurses must improvise when checking their patients’ feet.


*“We do not have any equipment for such assessments. Sometimes, if we really must check, we use the needle to see if they feel pain, which we don’t normally do, but in most cases, we don’t check their feet.”*



*“We do not have a kit, but with experience, we are trained to use the smallest of the needles. Sometimes you are scared to do this in case you injure the patient, and they don’t feel it.”*



*“A needle prick just to prove peripheral neuropathy, we don’t have kits. We just do basically use the needle and then the cotton wool, you run the cotton wool and can see if the patient can feel it.”*


## 4. Discussion

Diabetic foot is a significant complication of diabetes, frequently leading to infection, amputation, and increased mortality rates. It is estimated that between 19% and 34% of individuals with diabetes develop foot ulcers, with up to 60% of these cases progressing to infection. Among those experiencing moderate to severe infections, 20% undergo lower limb amputation [38]. The prognosis remains poor, as over half of the patients do not survive beyond five years post-ulceration [39], and the three-year mortality rate can reach 70% following amputation [40]. Despite these outcomes, early detection and intervention can substantially reduce the morbidity and mortality rates. Nevertheless, delayed care-seeking is prevalent [41], underscoring the need for regular foot screening.

In South Africa, approximately 80% of diabetes patients receive care at the PHC level [41]. Consequently, the early prevention of diabetic foot complications can be effectively managed within the primary healthcare (PHC) system, which is predominantly nurse-led. Their close relationship with patients and continuity of care position them at the forefront of diabetic foot care, facilitating the early detection and management of complications. However, this study identified five key challenges that limit their effectiveness: clinical knowledge gaps, systemic barriers, breakdowns in inter-facility communication, insufficient targeted training, and resource limitations.

### 4.1. Clinical Knowledge Gaps

One major concern was the insufficient knowledge and application of standardised tools for assessing diabetic foot risk and classifying ulcers. Although nurses recognised the significance of performing foot evaluations, many expressed a lack of confidence and often depended on subjective signs such as odour, visible necrosis, or patient complaints. Mafusi et al. (2024) observed that nurses exhibited inconsistent practices and had a limited grasp of diabetic foot care protocols, underscoring the widespread nature of this issue [7]. Without a solid understanding of validated tools such as the Wagner, PEDIS, or SINBAD classification systems, nurses are less prepared to accurately evaluate risk, prioritise care, or make timely referrals. This situation undermines patient outcomes and perpetuates a cycle of uncertainty and reactive care rather than fostering proactive, evidence-based practice at this level of care.

Awareness of globally recognised classification systems such as Wagner, PEDIS, and SINBAD was limited. Each of these systems offers distinct benefits: Wagner is easy to use but lacks detailed clinical information; PEDIS provides a thorough, multidimensional evaluation; and SINBAD is tailored for low-resource environments, making it particularly appropriate for primary healthcare settings [42]. A recent review by Monteiro-Soares et al. (2023) suggested using SINBAD for communication and audits in low-resource areas because of its simplicity and dependability [43]. Nonetheless, these tools are not widely used by nurses.

The underutilisation of these tools in PHC settings underscores a significant training gap and the broader marginalisation of diabetic foot care in routine practice in the primary healthcare settings. This observation aligns with the global literature, which suggests that nurses often rely on informal learning due to limited access to structured education, leading to variability in the quality of care [20,44]. Targeted training and integration of podiatric expertise into PHC are essential to enhance assessment accuracy and patient outcomes. Given the strong link between competence and confidence, this approach could empower nurses and boost their confidence in managing patients with diabetic foot complications [45]. Conducting structured foot assessments at an early stage is crucial for identifying complications before they become more severe. Enhancing nurses’ capabilities through targeted training and comprehensive foot health programmes offers a practical approach to improving patient outcomes and mitigating the impact of diabetic foot disease in primary care settings.

### 4.2. Systemic Barriers to Effective Care

Structural constraints within the healthcare system, particularly time limitations and excessive workloads, have emerged as significant barriers to effective diabetic foot care. Nurses consistently report that high patient volumes and competing clinical demands severely limit their ability to perform thorough foot assessments. This challenge is also evident in a Johannesburg-based study, which found that nurses frequently relegated foot care to a lower priority because of overwhelming caseloads and inadequate support. Nurses reported that high patient volumes, often exceeding 40 consultations per day, leave them with limited time for thorough evaluations. These findings align with previous research, indicating that an increased patient load significantly reduces consultation time in primary healthcare settings [46]. Participants in this study confirmed that the pressure to meet daily quotas, often up to 40 consultations, limits nurses’ ability to provide comprehensive care [47,48].

These systemic pressures contribute to the perception that primary healthcare nurses inadequately manage diabetic foot complications in patients. Rather than reflecting individual shortcomings, such perceptions often stem from institutional challenges that constrain nurses’ ability to provide optimal care [49,50]. This understanding helps to dismiss anecdotal and empirical criticisms, which may be attributed to the systemic challenges nurses face [51]. Addressing these underlying issues can improve the quality of care and outcomes for patients at risk of developing diabetic foot complications.

### 4.3. Breakdown in Interfacility Communication

A significant systemic barrier to effective diabetic foot care identified in this study was the lack of structured communication between primary and tertiary healthcare professionals. This issue disrupts continuity of care and undermines clinical decision-making. The absence of reliable feedback mechanisms following patient referrals from primary healthcare to tertiary facilities significantly hampers coordinated management. Feedback from tertiary facilities is crucial for appropriate follow-up and comprehensive primary healthcare [52]. Receiving feedback from tertiary facilities after patient referrals is vital for ensuring proper follow-up and continuity of care at the PHC level. Research underscores the importance of a collaborative and ongoing integrated care approach for patients with diabetic foot complications, as it can significantly improve their health outcomes and overall quality of life [53]. The absence of this element can negatively impact patient outcomes and impede the professional development of PHC nurses.

This study revealed that nurses seldom received feedback after referring patients with diabetic foot complications. Typically, they only learn about amputations when patients return for follow-up appointments without any updates on the hospital’s management or outcomes. The lack of feedback not only jeopardises patient outcomes but also limits opportunities for learning and professional growth among primary healthcare staff. These findings align with previous research, which has indicated that primary healthcare nurses and doctors perceive the referral system as disorganised, ineffective, and at times harmful to patient care [48,54].

### 4.4. The Need for Training

A lack of specialised training in diabetic foot care poses a significant systemic barrier to effective management in primary healthcare (PHC) settings. Diabetic foot assessment is crucial for risk stratification and individualised care planning, necessitating the evaluation of pulses, nerve function, ulcer history, deformities, callus formation, and self-care ability [55]. Despite its importance, the nurses in this study expressed frustration over their limited capacity to conduct comprehensive assessments, citing insufficient training as a primary constraint.

Although general wound care is part of basic nursing education, the complexity of diabetic foot pathology requires more targeted and context-specific instruction. Without this training, nurses often feel ill-equipped and hesitant to manage diabetic foot cases, leading to delayed referrals and poorer patient outcomes. Research indicates that this lack of confidence can impede informed decision-making, resulting in delayed referrals and compromised patient outcomes [56]. A recent local investigation revealed that fewer than 60% of nurses demonstrated average knowledge of diabetic foot ulcer risk factors and care protocols [16]. This was evident in this study, as nurses’ experiences underscore a critical gap in professional development and the need for ongoing, context-specific training tailored to PHC realities.

### 4.5. Limited Resources

The absence of dedicated foot health services and essential clinical tools in PHC settings presents a significant systemic obstacle to effective diabetic foot care in low-to middle-income countries (LMICs). The nursess in this study highlighted a critical gap in service delivery due to limited access to podiatristss, particularly in remote areas. Unlike other specialised services, such as oral, eye, or mental healthcare, foot health is often deprioritised, leaving nurses to manage complex cases that exceed their training and scope of practice. Handling these intricate diabetic foot cases poses a considerable challenge for nurses, often extending beyond their formal training and the scope of practice. Evidence suggests that optimal foot care in primary healthcare settings requires collaborative support from a multidisciplinary team, including podiatrists, physicians, and specialists [57,58].

The challenges nurses face are further exacerbated by the lack of basic diagnostic equipment such as monofilaments, Doppler devices, and foot screening kits. The absence of these tools limits nurses’ ability to conduct thorough assessments and undermines the quality of care. Participants expressed frustration at having to manage diabetic foot complications without specialist input, often feeling isolated and underprepared to do so.

The lack of podiatric integration reflects a broader systemic oversight in recognising foot health as a vital component of chronic disease management. To enhance diabetic foot care in PHC, nurses advocated for a coordinated multidisciplinary approach that is essential to improve patient outcomes. Nurses, podiatrists, and physicians must work collaboratively to ensure early detection, appropriate management, and timely referral. Integrating podiatry into PHC, along with targeted training and adequate resources, will strengthen clinical decision-making, boost nurses’ confidence, and reduce the burden of diabetic foot complications.

## 5. Conclusions

This study highlights critical gaps in diabetic foot care within South Africa’s primary PHC system. Nurses face challenges such as limited clinical knowledge, inadequate training, and resource shortages, which hinder the early detection and effective management of diabetic foot complications. Their experiences also reflect emotional strain and a disconnect between policy expectations and frontline realities.

Empowering nurses through targeted practical training is essential. Programmes should focus on diabetic foot risk stratification and ulcer classification, using internationally recognised tools such as the IWGDF and ADA guidelines, along with the Wagner, PEDIS, and SINBAD systems. These tools are vital for standardising care, improving referral accuracy, and building clinical confidence.

A shift from reactive to proactive care requires a well-resourced, integrated approach. Incorporating podiatrists into PHC—through permanent posts or rotational outreach—would provide specialised support. Structured foot health services with clear referral pathways and defined roles for nurses, podiatrists, and physicians are key to reducing fragmentation and promoting continuity of care.

### 5.1. Implications for Practice

Immediate policy action is required to address systemic gaps in diabetic foot care. Without intervention, patients remain at risk of preventable complications, and frontline healthcare workers continue to operate under strain. To ensure sustainable improvements in diabetic foot care, policy responses must be context-sensitive, evidence-informed, and aligned with the operational realities of PHC settings:Mandate diabetic foot care training for PHC nurses.Integrate podiatry services into PHC through permanent or rotational placements.Establish clear referral pathways and team-based care protocols.Ensure availability of essential equipment and resources at PHC facilities.Introduce multidisciplinary teams at the PHC level, including nurses, podiatrists, and physicians, to support collaborative, patient-centred diabetic foot care.

### 5.2. Transferability

These findings are applicable to other low- and middle-income countries with similar PHC structures and resource limitations. Health systems facing comparable challenges can adapt these strategies to strengthen diabetic foot care, improve patient outcomes, and foster multidisciplinary collaboration at the primary level.

## Figures and Tables

**Table 1 ijerph-22-01799-t001:** Participants’ characteristics (N = 21).

Variable	Type	Frequency	%
Age	20–25	2	9%
	26–35	5	24%
	36–45	4	19%
	46–55	6	29%
	56	4	19%
Sex	Male	3	14.3%
	Females	15	71.4%
	Prefer not to say	3	14.3%
Qualification	Diploma in Nursing	7	33.3%
	Bachelor of Nursing	12	55.6%
	Post-graduate	2	11.1%
	Training in diabetic foot assessment	0	0%
Working experience	≥5 year	36	6%
	6–10 years	269	45%
	11–15 years	96	16%
	16–20 years	88	15%
	20+ years	30	5%

**Table 2 ijerph-22-01799-t002:** Emerging themes and subthemes.

Theme	Subtheme
Knowledge of diabetic foot ulcers	Clinical knowledge gaps
Time and Workload Constraints	Competing demands in clinical workflow
Poor or non-existent feedback from hospitals	Breakdown in interfacility communication
Need for targeted training	Training gaps in wound care management
Resource Limitations	Systemic resource constraints Limited access to assessment technologies

## Data Availability

Data will be made available on reasonable request following the receipt of authorisation from the author’s local ethics committee.

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
