# Peer review of "A Phenomenological Study of Primary Healthcare Nurses’ Experiences in Assessing and Managing Diabetic Foot Complications"

_ijerph, 2025, doi:10.3390/ijerph22121799_

Round 1

Reviewer 1 Report

Comments and Suggestions for Authors

1- The title is clear and descriptive, accurately reflecting the study’s focus on nurses’ experiences managing diabetic foot care in PHC settings.

2- The keywords are relevant but could be improved by using concise, standardized terms (like MeSH) to enhance clarity and discoverability. Consider shortening long phrases and including core concepts such as “Diabetic Foot,” “Nurses,” and “Primary Health Care” for better indexing.

3-In the abstract, the objective needs to be clarified. Also, while the results focus on nurses’ experiences and challenges in managing diabetic foot complications, the conclusion shifts to discussing “awareness,” which isn’t clearly supported by the data. 
To strengthen the abstract, revise the conclusion to directly reflect the main findings. For example, emphasize the specific challenges and needs identified in the results, such as training gaps or resource limitations, rather than introducing new ideas like awareness. This will provide a clearer, more coherent summary that better represents the study’s outcomes.

4-For the introduction: it is overly long, with repetitive statements regarding nurses’ roles, prevalence data, and the burden of diabetic foot complications. This dilutes the central message and affects readability. Also, it lacks the international context of the literature. A clear statement of the aim at the end of the study is essential. 

5- For the method, provide more information about the details of data collection to improve the reproducibility of the study.

6- The results could be enriched by adding subheadings to the main themes. For example,  subthemes  for Time and Workload Constraints, with providing quotes for each one.

7- Conclusion: Avoid introducing new ideas, and summarize the main ideas, linking them with national and international studies.  

Reviewer 2 Report

Comments and Suggestions for Authors

The topic is highly relevant and addresses a significant public health issue in South Africa, where diabetic foot complications remain a major burden. The phenomenological methodology is well-justified and executed with appropriate rigor (COREQ, SRQR, Colaizzi’s method). Ethical approval and participant confidentiality are clearly addressed. The thematic analysis (five main themes: limited knowledge, workload/time constraints, lack of feedback, need for training, resource shortages) is coherent and provides valuable insights. The manuscript also makes strong connections between findings and implications for health policy and practice, especially regarding nurse education and podiatry integration.

1) The demographic breakdown is limited (e.g., age range, training level, work setting). Providing more detail would enhance contextual interpretation.

2) Some verbatim quotes are lengthy and repetitive; these could be shortened to improve clarity and strengthen the analytic narrative.

3) There appears to be a reporting error (mean age reported as 4.6 years; nursing experience 4 years). Please correct this.

4) The discussion references IWGDF and ADA guidelines briefly, but a more detailed comparison with international standards (e.g., Wagner, PEDIS, SINBAD classification systems) is recommended.

5) The limitations section should be expanded. Geographic restriction (Gauteng only), limited generalizability, and possible bias from self-reported experiences should be acknowledged more explicitly.

6) Emphasize multidisciplinary collaboration (nurses, podiatrists, physicians) more strongly to highlight the practical implications of your findings.

Comments on the Quality of English Language

The English language of the manuscript is clear and professional. Minor stylistic improvements could be made, but overall, the text is fluent and does not hinder comprehension.

Reviewer 3 Report

Comments and Suggestions for Authors

First, I would like to thank the author for the opportunity to review the article "A Phenomenological Study of Primary Health Care Nurses' Experiences in Assessing and Managing Diabetic Foot Complications" and congratulate the author on the adequacy of the topic. I would like to make a number of recommendations that I believe could improve the article:
The abstract could benefit from synthesizing different phrases, indicating the characteristics of the nurses selected for the study (community health clinics, etc.), and mentioning the use of COREQ and SRQR in the article.
The introduction justifies the need for the work, although I recommend a summary of those aspects or ideas that may be repeated, such as the need to explore experiences or the lack of protocols.
The methodology is clear and uses COREQ and SRQR criteria, which facilitate transparency. The explanation and detail of the two-stage sampling and the inclusion of bracketing and reflexivity are appreciated. However, there are details that should be clarified, including when sample saturation was considered, and a description of the analysis process (initial codes, triangulation, etc.).
In the results, I assume that in Table 1, the mean age indicated is a typo, and the value indicated in Table 1 (Mean: 4.6) is incorrect. Regarding the table, it could be improved by indicating the sample characteristics (years of experience, district, whether the patient has received specific training, etc.). Some testimonials that repeat similar ideas should be clarified; this could be summarized to avoid redundancies.
The discussion occasionally repeats some of the results and does not include possible limitations of the study, such as the fact that the study was conducted in a single district, methodological limitations, etc. At the same time, I would recommend including a section on implications for practice, indicating that the importance of including podiatry professionals in the healthcare system could be discussed.
Regarding the conclusions, I would recommend summarizing them in shorter sentences, indicating the results of the study and the prioritization of recommendations over different timeframes, as well as the potential transferability of the study to similar healthcare systems.
Regarding the references, it is recommended to review any incomplete references, such as references 4 and 18. Two self-citations have been seen (8 and 14), but they are relevant to the topic.

Round 2

Reviewer 3 Report

Comments and Suggestions for Authors

The authors have modified the key aspects indicated in the review and I consider that the article meets the minimum criteria for publication.

Author Response

The authors have modified the key aspects indicated in the review and I consider that the article meets the minimum criteria for publication.

Noted with appreciation. I have also addressed the comments from the editor.